# Atrial Substrate Underlies the Recurrence after Catheter Ablation in Patients with Atrial Fibrillation

**DOI:** 10.3390/jcm9103164

**Published:** 2020-09-30

**Authors:** Yong-Soo Baek, Jong-Il Choi, Yun Gi Kim, Kwang-No Lee, Seung-Young Roh, Jinhee Ahn, Dong-Hyeok Kim, Dae In Lee, Sung Ho Hwang, Jaemin Shim, Jin Seok Kim, Dae-Hyeok Kim, Sang-Weon Park, Young-Hoon Kim

**Affiliations:** 1Division of Cardiology, Department of Internal Medicine, Inha University College of Medicine and Inha University Hospital, Incheon 22212, Korea; existsoo@inha.ac.kr (Y.-S.B.); kdhmd@inha.ac.kr (D.-H.K.); 2Division of Cardiology, Department of Internal Medicine, Korea University College of Medicine and Korea University Medical Center, Seoul 02841, Korea; tmod0176@gmail.com (Y.G.K.); knlee81@gmail.com (K.-N.L.); rsy008@gmail.com (S.-Y.R.); reinee81@naver.com (J.A.); tomas9912@naver.com (D.-H.K.); acttopia@naver.com (D.I.L.); jshim@korea.ac.kr (J.S.); heartmania@unitel.co.kr (J.S.K.); swparkmd@gmail.com (S.-W.P.); yhkmd@unitel.co.kr (Y.-H.K.); 3Department of Radiology, Korea University Anam Hospital, Seoul 02841, Korea; sungho77@korea.ac.kr

**Keywords:** atrial fibrillation, catheter ablation, substrate, magnetic resonance imaging

## Abstract

Prediction of recurrences after catheter ablation of atrial fibrillation (AF) remains challenging. We sought to investigate the long-term outcomes after AF catheter ablation. A total of 2221 consecutive patients who underwent catheter ablation for symptomatic AF were included in this study (mean age 55 ± 11 years, 20.3% women, and 59.0% paroxysmal AF). Extensive ablation, in addition to circumferential pulmonary vein isolation, was more often accomplished in patients with non-paroxysmal AF than in those with paroxysmal AF (87.4% vs. 25.3%, *p* < 0.001). During a median follow-up of 54 months, sinus rhythm (SR) was maintained in 67.1% after index procedure. After redo procedures in 418 patients, 83.3% exhibited SR maintenance. Recurrence rates were similar for single and multiple procedures (17.4% vs. 16.7%, *p* = 0.765). Subanalysis showed that the extent of late gadolinium enhancement (LGE), as assessed by cardiac magnetic resonance, is greater in patients with recurrence than in those without recurrence (36.2 ± 23.9% vs. 21.8 ± 13.7%, *p* < 0.001). Cox-regression analysis revealed that non-paroxysmal AF (hazard ratio (HR) 2.238, 95% confidence interval (CI) 1.905–2.629, *p* < 0.001), overweight (HR 1.314, 95% CI 1.107–1.559, *p* = 0.020), left atrium dimension ≥ 45 mm (HR 1.284, 95% CI 1.085–1.518, *p* = 0.004), AF duration (HR 1.020 per year, 95% CI 1.006–1.034, *p* = 0.004), and LGE ≥ 25% (HR 1.726, 95% CI 1.330–2.239, *p* < 0.001) are significantly associated with AF recurrence after catheter ablation. This study showed that repeated catheter ablation improves the clinical outcomes of patients with non-paroxysmal AF, suggesting that AF substrate based on LGE may underpin the mechanism of recurrence after catheter ablation.

## 1. Introduction

Atrial fibrillation (AF) is associated with higher rates of mortality, stroke, congestive heart failure, and hospitalization [1]. Given the limited efficacies and potential adverse effects of antiarrhythmic drugs, catheter ablation offers the most effective rhythm control strategy and could be utilized as a first-line treatment in selected patients [2,3]. Reports indicate that in patients with AF progression, multiple procedures may be more effective than single procedures in the long-term [4]. Atrial fibrosis is a major determinant of AF progression; thus, more extensive atrium remodeling favors development of persistent arrhythmia [5]. Atrial tissue fibrosis, which is a marker of an underlying substrate, could be visualized with late gadolinium enhancement (LGE) on cardiac magnetic resonance (MR) images and has been shown to be independently associated with the likelihood of recurrent arrhythmia [6]. However, clinical implications of such an association are challenging, and little data are available on the effect of catheter ablation on long-term clinical outcomes with respect to AF progression stratified by AF type and atrial fibrosis as estimated by MR LGE. In this study, we investigated the clinical outcomes of catheter ablation over a 10-year follow-up period. We hypothesized that atrial substrates represented by AF progression underlie the mechanism of recurrence after catheter ablation in patients with AF.

## 2. Materials and Methods

### 2.1. Study Population

A total of 2221 patients who underwent radiofrequency catheter ablation for symptomatic drug-refractory AF between 2004 and January 2016 at a single tertiary hospital were enrolled in this retrospective study. The study flow chart is presented schematically in Figure 1. The exclusion criteria applied were as follows: (1) AF with rheumatic valvular disease; (2) AF refractory to electrical cardioversion; and (3) prior cardiac surgery. In all patients, antiarrhythmic drugs were discontinued for a period of at least five half-lives prior to catheter ablation, and amiodarone was discontinued for at least 4 weeks prior to the procedure. In this study, AF was defined as paroxysmal if the episodes lasted <48 h, or terminated spontaneously within 7 days, or if the patients received electric or pharmacological cardioversion within 48 h. AF was defined as non-paroxysmal when AF progressed to persistent (sustained AF (i.e., >7 days), or when AF was successfully restored by electrical or pharmacologic cardioversion after 48 h), longstanding persistent (continuous AF for >12 months), or permanent AF (a decision was made not to restore or maintain sinus rhythm (SR) by any means). The study protocol was approved by the Institutional Review Board of Korea University Anam Hospital Institutional Review Board (2016-AN0210) and complied with the Declaration of Helsinki.

### 2.2. Electrophysiological Studies and Catheter Ablation

Ablation was performed using radiofrequency energy under three-dimensional (3D) electroanatomical mapping (NavX, St. Jude Medical, Minnetonka, MN, USA; CARTO3, Johnson & Johnson Inc., Diamond Bar, CA, USA) merged with 3D spiral computed tomography (CT) or cardiac magnetic resonance (CMR). Patients were sedated with intravenous midazolam and fentanyl, and arterial blood gas analysis was performed hourly. Two transseptal punctures were performed using long sheaths (Fast-Cath™ and Swartz™ SL1 or 2, St. Jude Medical, AF Division, Minnetonka, MN, USA) and a Brockenbrough needle under fluoroscopic guidance. Unfractionated heparin was administered to maintain an activated clotting time between 300 and 350 s. Radiofrequency energy was delivered at a maximum temperature of 48 °C and a power of 25–35 W using a 3.5-mm open irrigated-tip ablation catheter. All patients underwent circumferential pulmonary vein isolation (CPVI). The ablation strategy used was as previously described [7]. During catheter ablation for paroxysmal AF, triggers of sustained AF were evaluated and targeted after CPVI. To identify the triggers initiating paroxysmal AF, multiple electrical cardioversions were subsequently performed in patients with initial AF rhythm under isoproterenol (10–20 µg/min) infusion. In patients with SR, AF was induced by burst high right atrial pacing in decrements from 250 ms to the atrial refractory period. Triggers were identified based on reinitiation of sustained or non-sustained AF within 2 min of cardioversion. A single premature atrial beat without sustained AF plus atrial tachycardia (AT) was not considered a trigger. During the protocol, a 20-pole circular mapping catheter, an ablation catheter, and a quadripolar catheter were positioned in the left pulmonary vein, right pulmonary vein, and superior vena cava, respectively.

The first beat initiating AF was considered a trigger and the site of origin was analyzed. After CPVI, we repeated the same protocol at least three times to identify non-pulmonary vein focal initiators of AF. The endpoint of the procedure for paroxysmal AF was the elimination of all triggers, including those of the pulmonary vein (PV). For patients with persistent AF, 3D automated complex fractionated atrial electrography (CFAE) and/or linear ablation, including the roof line, posterior inferior line, perimitral isthmus line, and/or anterior line, was performed at the surgeons’ discretion. The endpoint of the procedure for persistent AF was AF termination and no induction of sustained atrial arrhythmia. CFAE map settings were as follows: refractory period 49 ms, P–P sensitivity >0.1 mV, and duration 30 ms. If ablation was performed, a CFAE map was used for guidance until AF was terminated or converted to AT or until fractionated activity was eliminated. Areas in the right atrium (RA) based on CFAE were targeted for ablation if AF persisted after extensive left atrium (LA) ablation. If AF was sustained after all RA CFAE had been abolished, a tailored approach was applied on the basis of case-by-case incidence that included LA and/or RA potential-guided focal or linear ablation. If AF was converted to AT during pulmonary vein isolation (PVI) or CFAE-guided ablation, activation and entrainment mapping-guided ablations were performed. Cavotricuspid isthmus (CTI) linear ablation and bidirectional block were performed in patients with clinical atrial flutter. In all patients with persistent AF, the inducibility of sustained AF plus AT by pacing was evaluated when AF and AT were terminated successfully by ablation. If SR was not restored after ablation of the aforementioned lesions, the patients were cardioverted using an internal or external direct-current electrical shock.

### 2.3. Post-Ablation Management and Follow-Up

Ambulatory monitoring and daily 12-lead electrocardiogram (ECG) were applied to patients at 1 to 2 days after AF ablation. Patients were asked to visit our outpatient clinic 1, 3, 6, and 12 months after catheter ablation and every 6 months thereafter, or whenever symptoms occurred. Oral anticoagulant and antiarrhythmic medications were continued for at least 3 months after the procedure. The decision to discontinue antiarrhythmic medications after 3 months was made at the physicians’ discretion. All patients underwent electrocardiography at every follow-up visit and a 24-h Holter recording was obtained at 3 and 6 months and every 6 months thereafter. However, whenever patients reported palpitations, Holter monitor or event monitor recordings were obtained and evaluated to check for arrhythmia recurrence. We defined AF recurrence as any episode of AF plus AT lasting for ≥30 s. AF recurrence based on electrocardiography after a blanking period of 3 months (i.e., at ≥3 months post-ablation) was documented as clinical recurrence.

### 2.4. Analysis of MR LGE Images

Cardiac MR examinations were performed using a 3-T MR system (Achieva; Philips Medical Systems, Best, The Netherlands) equipped with a 32-element phased-array cardiac coil. All cardiac MR LGE images were analyzed independently by two experienced radiologists (blinded to the patients’ clinical and electrophysiological data) at a commercial software workstation (Terarecon iNtuition; TeraRecon, Foster City, CA, USA). The 3D reconstruction process used to produce MR LGE images was as previously described [8,9]. On transverse or coronal MR LGE images, epicardial and endocardial borders of LA walls were semi-automatically contoured to select the entire LA walls and reconstruct 3D LA models. For quantitative analysis of a selected LA wall, we used signal threshold methods and manual delineation of regions of interest (ROIs) that included, to the greatest possible extent, the normal left ventricular (LV) wall of nulled signal intensity on an MR LGE image. Six and two standard deviation (SD) thresholds above the mean signals of normal LV walls were calculated using ROI means and SD values. Using the six and two SD thresholds, LA wall substrates were classified into three groups: (1) fibrotic substrate with a signal ≥6 SD threshold; (2) intermediate substrate with a signal ≥2 SD threshold but <6 SD threshold; and (3) normal substrate with a signal <2 SD threshold. In addition, to better delineate LA wall composition, colored look-up table masks were applied to 3D LA models as follows: yellow for fibrotic substrate, gray for indeterminate substrate, and blue for normal substrate. Figure 2 shows two examples of processed 3D reconstructions of cases with low and high MR LGE at LA walls.

### 2.5. Statistical Analysis

Results are presented as means ± SD for continuous variables and as proportions for categorical variables. For patients with or without clinical recurrence, continuous variables were compared using Student’s *t*-test, and categorical variables were analyzed using the chi-square test. The *p* values less than 0.05 were regarded as statistically noticeable in our study. Times to AF plus AT recurrence were calculated by Kaplan–Meier analysis, and comparisons were performed using log-rank statistics. Univariate regression analysis was used to identify relationships between clinical variables and AF/AT recurrences after index ablation. Multivariate logistic regression analysis was performed using variables with *p* values <0.05 in the univariate analysis to identify predictors of clinical recurrence following ablation. Statistical analysis was performed using SPSS software version 24.0 (IBM, Armonk, NY, USA).

## 3. Results

### 3.1. Patient Characteristics

Baseline patient characteristics are presented in Table 1. The mean age of the 2221 subjects was 55.1 ± 10.8 years, and 20.3% were female. The mean antero-posterior LA diameter and LV ejection fraction, which were determined using echocardiographic findings, were 41.2 ± 6.1 mm and 55.1 ± 5.8%, respectively (Table 2).

The proportions of non-paroxysmal AF (*p* < 0.001), heart failure (*p* = 0.005), hypertension (*p* = 0.021), and diabetes mellitus (*p* = 0.047) were significantly greater in patients with clinical recurrence than in those without. Preprocedural echocardiograms showed that mean LA dimension was larger (*p* < 0.001) and mean LV ejection fraction was lower (*p* < 0.001) in patients with recurrence. Furthermore, mean AF duration (*p* < 0.001), mean CHA_2_DS_2_-VASc score (p < 0.001), and mean body weight (*p* = 0.04) were greater in patients with clinical recurrence.

### 3.2. Radiofrequency Catheter Ablation: Index and Repeat Procedures

All CPVIs were successfully achieved during the first procedure. A total of 1425 patients (64.0%) underwent CTI ablation with bidirectional block. CFAE and linear ablation were performed in 507 (22.8%) and 268 (12.1%) patients, respectively, and 520 (23.0%) patients had biatrial ablation in the LA and RA, as previously described. Clinical outcomes after ablation are summarized in Figure 1. During a median follow-up period of 54 months after the index procedure, SR was maintained in 1490 of the 2221 study subjects (67.1%). Of the 731 patients diagnosed with clinical recurrence after the index procedure, 416 (56.9%) had recurrence of AF or atrial flutter and 313 (42.8%) had AT recurrence. Second, third, and fourth procedures were performed in 418, 70, and 4 patients, respectively, of the 731 patients, and the corresponding rates of SR maintenance were 83.3%, 69.6%, and 33.3% after the repeat procedures. The mean procedure time for the 2221 study subjects was 275 ± 108 min, the mean ablation time was 114 ± 54 min, and the mean total ablation and procedure times were 114 ± 54 and 275 ± 108 min, respectively (Table 2).

### 3.3. Relationship between MR LGE and Recurrence

Representative MR LGE findings in patients with or without recurrence are provided in Figure 2. Preprocedural MR imaging was performed to evaluate the LA and PVs in 405 patients before catheter ablation. Patients who experienced recurrence had a significantly greater mean LGE area percentage than those who did not experience recurrence (36.2 ± 23.9% vs. 21.8 ± 13.7%, *p* < 0.001) (Figure 3).

### 3.4. Predictors of Clinical Recurrence

Univariate and multivariate analyzes were performed to identify predictors of clinical recurrence (Table 3). Multivariate model 1, which was adjusted for relevant baseline risks, showed that non-paroxysmal AF (hazard ratio (HR) 2.254, 95% confidence interval (CI) 1.919–2.648, *p* < 0.001), overweight/obesity (HR 1.391, 95% CI 1.176–1.937, *p* = 0.018), LA dimension ≥45 mm (HR 1.279, 95% CI 1.005–1.033, *p* = 0.004), and AF duration (HR 1.019, 95% CI 1.005–1.033, *p* = 0.008) are independently associated with clinical recurrence. In multivariate model 2, which included MR LGE and was adjusted for relevant baseline risks, an MR LGE area percentage ≥25% independently predicted clinical recurrence (HR 1.726, 95% CI 1.330–2.239, *p* < 0.001).

### 3.5. Effect of Multiple Procedures According to AF Type

During a mean follow-up of 57.2 ± 37.9 months, SR was maintained in 76.6% of patients with paroxysmal AF and in 53.5% of those with non-paroxysmal AF (log rank *p* < 0.001, Figure 4A). Kaplan–Meier estimates of AF plus AT-free survival rates showed that the recurrence rates were similar between patients who underwent single and those who had multiple procedures (log rank *p* = 0.863, Figure 4B). Kaplan–Meier analyzes according to AF type and number of procedures showed that multiple and single procedures have similar AF plus AT-free survival rates in patients with paroxysmal AF (log rank *p* = 0.661). Patients with non-paroxysmal AF who received multiple procedures had a significantly lower recurrence rate after the last procedure than those who underwent a single procedure (log rank *p* = 0.022) (Figure 5).

## 4. Discussion

### 4.1. Main Findings

This study provides long-term clinical outcome data (over a 10-year follow-up period) of patients with AF who underwent catheter ablation. The main findings of the study are as follows: (1) patients who experienced recurrence were more likely to have hypertension, diabetes, a higher CHA_2_DS_2_-VASc score, and heart failure; (2) multiple and single procedures have similar recurrence rates in patients with paroxysmal AF, and multiple procedures have better clinical outcomes than single procedures in those with non-paroxysmal AF; (3) recurrence rates were significantly higher in patients with non-paroxysmal AF during the long-term follow-up; (4) cardiac MR data showed that patients with recurrence had increased LGE; and (5) LGE ≥25%, non-paroxysmal AF, overweight/obesity, and AF duration were associated with clinical recurrence.

### 4.2. Long-Term Clinical Outcomes According to AF Type

Ouyang et al. reported a stable SR in 46.6% of patients after a single CPVI procedure and in 79.5% of patients with paroxysmal AF after multiple procedures, and progression toward chronic AF in only 2.4% of the patients after a median follow-up of 4.6 years [10]. Previous studies have reported success rates of 56–58.7% after index CPVI in patients with paroxysmal AF, which increased to 87–89.4% after multiple procedures [3,11]. In patients with persistent AF, 59% of patients assigned to a PVI alone group were free of recurrent AF, whereas 49% of patients assigned to a PVI plus CFAE group and 46% of patients assigned to a PVI plus linear ablation group remained free of AF (*p* = 0.15) [12]. Scherr et al. investigated the long-term outcomes of 150 patients treated using a stepwise ablation approach for persistent AF and found arrhythmia-free survival rates of 35.3 ± 3.9, 28.0 ± 3.7, and 16.8 ± 3.2% at, respectively, 1, 2, and 5 years after a single procedure [13]. A meta-analysis demonstrated that a single AF procedure resulted in freedom from atrial arrhythmia over long-term follow-up in 53.1% (95% CI 46.2–60.0%) of all patients, 54.1% (95% CI 44.4–63.4%) of patients with paroxysmal AF, and 41.8% (95% CI 25.2–60.5%) of patients with non-paroxysmal AF. Moreover, the long-term overall success rate after multiple procedures was 79.8% (95% CI 75.0–83.8%) [4].

AF is a progressive disease associated with structural remodeling, which leads to a gradual increase in AF burden [2]. Recently, Tilz et al. reported a rate of progression to persistent AF of only 6.2% over a 10-year follow-up after catheter ablation and suggested that CPVI may delay or prevent AF progression [3]. According to our findings, patients with paroxysmal AF had better clinical outcomes than those with non-paroxysmal AF, and those with non-paroxysmal AF who underwent multiple procedures had a lower recurrence rate than those who had a single procedure. These findings strongly suggest that in non-paroxysmal AF, multiple procedures and early intervention (such as catheter ablation in paroxysmal AF) play an important role in preventing progression to the irreversible stage.

### 4.3. Clinical Predictors of SR Maintenance After Catheter Ablation

AF ablation is associated with various recurrence rates, mostly due to patient-specific preprocedural factors and specific procedural factors [14]. Predictors of recurrence after AF ablation divided depending on underling structural, electrical, and autonomic substrate [14,15]. In this study, age, non-paroxysmal AF, overweight/obesity, hypertension, diabetes mellitus, LA diameter, MR LGE, and AF duration were found to be associated with AF recurrence by univariate analysis. Previous studies suggested that multiple factors, including persistent AF, AF duration, LA diameter, presence of underlying cardiomyopathy, and previous episodes of congestive heart failure influence long-term follow-up efficacy of radiofrequency catheter ablation in AF [16]. Much evidence indicates that overweight/obesity is associated with the development of AF [17]. A high body mass index has also been shown to be associated with an increased risk of AF recurrence after catheter ablation and progression from paroxysmal to permanent AF [18], and it has also been reported that delayed AF recurrence after catheter ablation is associated with the risk of components of metabolic syndrome, such as hypertension, diabetes mellitus, and overweight status [19]. Atrial fibrosis is a condition that involves an underpinned underlying substrate and could be visualized on cardiac MR images, which could also be used to individualize catheter ablation strategies. In a previous study, atrial tissue fibrosis, as estimated by delayed enhancement on MR images, was reported to be independently associated with the risk of recurrent arrhythmia in patients with AF after catheter ablation [6].

### 4.4. Effect of Atrial Substrate on Clinical Recurrence

In this study, patients with AF recurrence after catheter ablation exhibited more LA remodeling and fibrosis based on echocardiography and MR LGE. In patients with AF, atrial fibrosis is a major determinant of the success of rhythm control strategies, including catheter ablation [6,20]. Delayed enhancement on MR images is a known marker of fibrotic non-viable ventricular myocardium [20], and atrial fibrosis stratified by MR LGE may aid in selection of appropriate candidates and optimal strategies for catheter ablation of AF [21]. Therapeutic approaches that utilize delayed-enhancement MR imaging for detection of left atrial fibrosis provide a means of enhancing the tailored management of AF [22]. In our study, we observed that an MR LGE ≥25% is strongly associated with clinical recurrence after AF ablation during long-term follow-up. This finding is consistent with the findings of the Delayed-Enhancement MRI Determinant of Successful Radiofrequency Catheter Ablation of Atrial Fibrillation (DECAAF) study, in which atrial fibrosis, as estimated by delayed enhancement on MR images, was found to be independently associated with the likelihood of recurrent arrhythmia [6]. MR LGE to identify the atrial substrate is important in two aspects: one is that it can avoid unnecessary AF ablation procedures, and the other is that it can be feasible and effective for targeting re-entrant drivers or anatomic MRI LGE detected gaps in the AF ablation strategy [23,24]. Therefore, our results indicate that atrial fibrosis, as estimated using MR LGE, may offer a robust means of determining optimal AF management and deciding upon preemptive strategies to prevent AF progression. In the longer term, efforts to identify atrial substrate using MR LGE would allow more patient-centered intervention and patient-tailored decision-making for better clinical outcome in patients undergoing AF ablation.

### 4.5. Study Limitations

This study has several limitations that warrant consideration. First, this study is inherently limited by its non-randomized, retrospective, single center registry-based design. Second, ECG monitoring and 24-h Holter ECG might have missed asymptomatic episodes of AF. Third, cardiac MR was performed in only 405 of the 2221 patients because it was conducted on consecutive patients from 2013. Fourth, we did not perform MR LGE after AF ablation to evaluate how the fibrotic substrate alters before and after the procedure.

## 5. Conclusions

In conclusion, this long-term follow-up study showed that catheter ablation is an effective strategy for rhythm control in patients with non-paroxysmal AF requiring multiple procedures and in those with paroxysmal AF. Atrial substrate, as estimated by MR LGE, is independently associated with clinical outcome after catheter ablation of AF, suggesting that it is an underlying factor of AF recurrence.

## Figures and Tables

**Figure 1 jcm-09-03164-f001:**
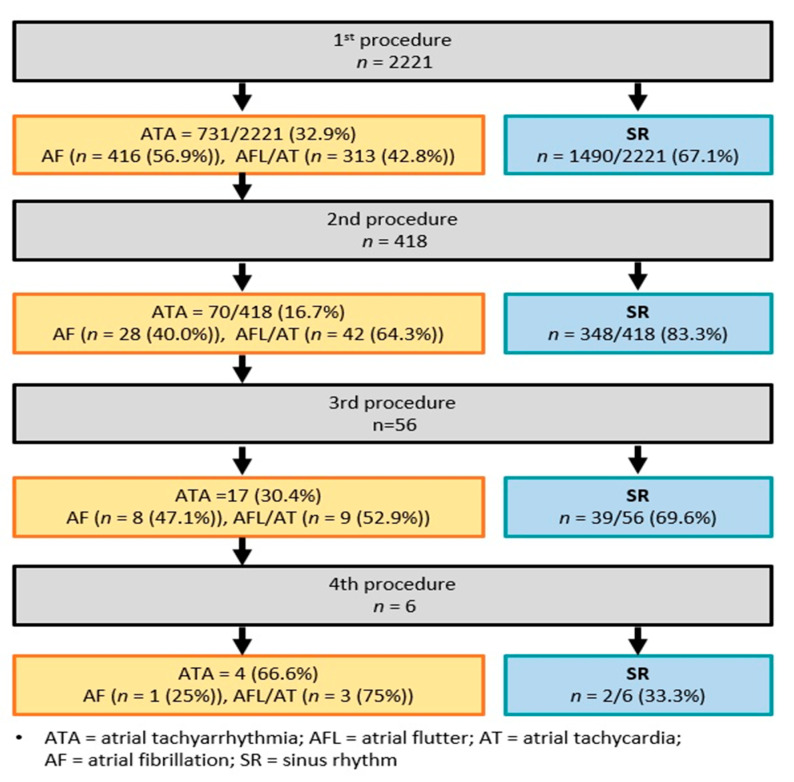
Flow chart of atrial fibrillation (AF) catheter ablation outcomes.

**Figure 2 jcm-09-03164-f002:**
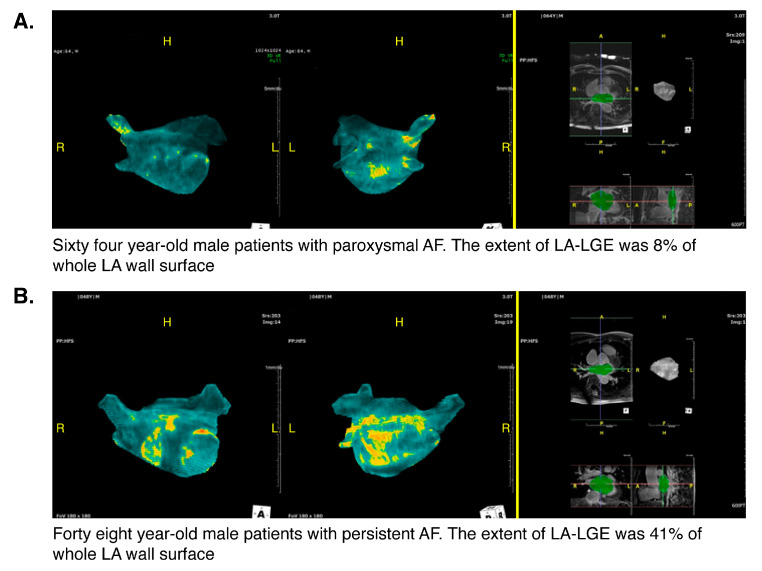
Processed 3D reconstructions showing low and high late gadolinium enhancement in cardiac magnetic resonance images (MR LGE) within left atrium (LA) walls. (**A**) A 64-year-old man with paroxysmal AF. (**B**) A 48-year-old man with persistent AF.

**Figure 3 jcm-09-03164-f003:**
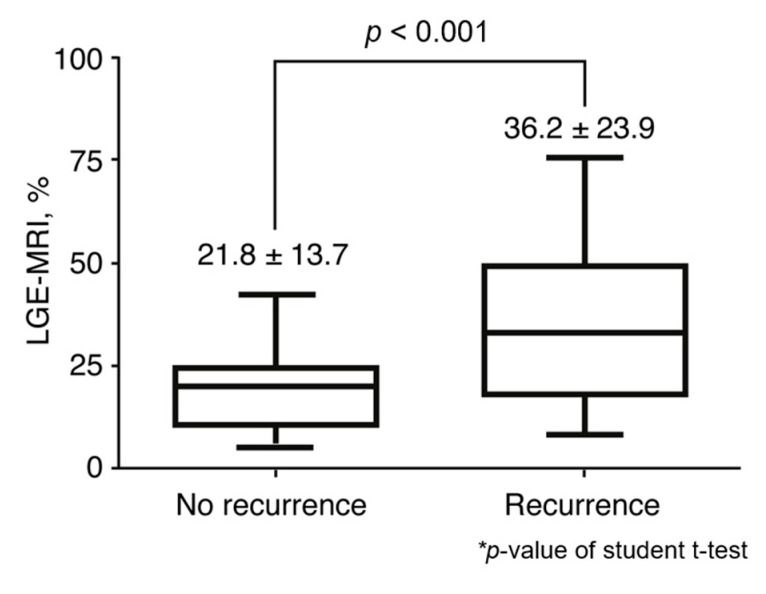
Comparison of MR LGE burdens between patients with and those without clinical recurrence after catheter ablation.

**Figure 4 jcm-09-03164-f004:**
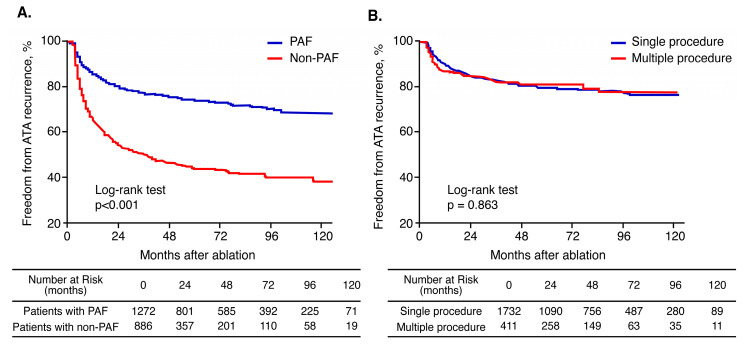
Kaplan–Meier curves showing (**A**) cumulative AF-free survival rates in patients with paroxysmal AF and non-paroxysmal AF and (**B**) AF-free survival rates in patients who underwent single and those who had multiple procedures.

**Figure 5 jcm-09-03164-f005:**
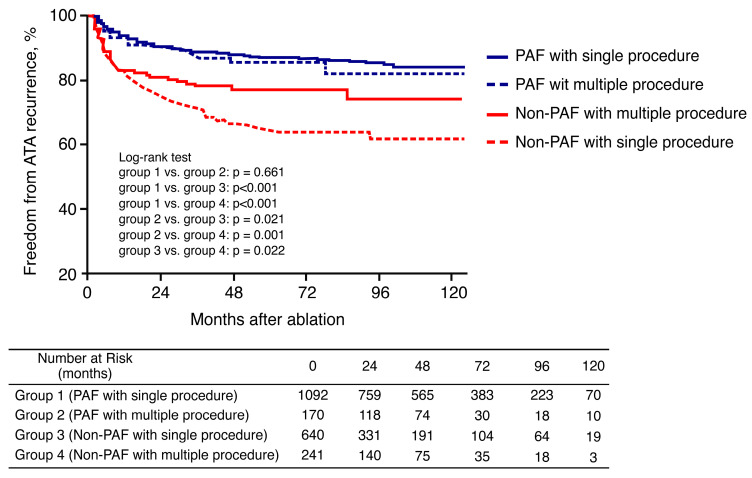
Kaplan–Meier curves according to AF type and number of procedures.

**Table 1 jcm-09-03164-t001:** Baseline clinical characteristics of study patients according to clinical recurrence after catheter ablation.

	Overall	No Recurrence Group	Recurrence Group	* *p* Value
	(*n* = 2221)	(*n* = 1490)	(*n* = 731)	
Age (years)	55.1 ± 10.8	55.0 ± 10.8	56.5 ± 10.7	0.004
Female, *n* (%)	451 (20.3)	303 (20.3)	148 (20.2)	0.961
Non-paroxysmal AF, *n* (%)	911 (41.0)	487 (32.7)	424 (58.0)	<0.001
Height, cm	168.2 ± 8.2	168.0 ± 8.0	168.5 ± 8.1	0.269
Weight, kg	70.8 ± 11.4	70.4 ± 11.6	71.7 ± 11.1	0.040
BSA, m^2^	1.8 ± 0.4	1.8 ± 0.5	1.8 ± 0.2	0.927
BMI, kg/m^2^	25.1 ± 3.4	24.9 ± 3.5	25.3 ± 3.2	0.078
HF, *n* (%)	320 (14.4)	193 (13.0)	127 (17.4)	0.005
Hypertension, *n* (%)	732 (33.0)	467 (31.3)	265 (36.3)	0.021
DM, *n* (%)	171 (7.7)	103 (6.9)	68 (9.3)	0.047
Stroke/TIA, *n* (%)	116 (5.2)	72 (4.8)	44 (6.0)	0.237
Vascular disease, *n* (%)	104 (4.7)	76 (5.1)	28 (3.8)	0.200
CHA_2_DS_2_-VASc score	1.15 ± 1.22	1.09 ± 1.20	1.28 ± 1.24	<0.001
Post-ABL medication				
Beta-blocker, *n* (%)	403 (18.4)	268 (18.2)	248 (34.3)	<0.001
Statin, *n* (%)	516 (23.5)	224 (15.2)	179 (24.8)	<0.001
AAD, *n* (%)	342 (15.6)	192 (13.0)	150 (20.7)	<0.001
Laboratory findings				
BUN (mg/dl)	15.7 ± 4.8	15.7 ± 4.9	15.6 ± 4.5	0.733
Creatinine (mg/dl)	1.0 ± 0.4	1.0 ± 0.4	1.0 ± 0.2	0.381
Total cholesterol (mg/dl)	179.7 ± 37.1	180.3 ± 35.9	177.8 ± 40.4	0.419
HDL cholesterol (mg/dl)	48.2 ± 11.8	49.3 ± 12.3	49.2 ± 12.2	0.843
LDL cholesterol (mg/dl)	108.0 ± 30.5	109.0 ± 30.4	105.9 ± 30.4	0.053
Triglyceride (mg/dl)	136.3 ± 83.9	136.3 ± 86.1	136.3 ± 79.1	0.996
HbA1c	5.9 ± 0.8	5.9 ± 0.8	5.9 ± 0.7	0.465
AF duration, years	4.6 ± 4.7	4.2 ± 4.4	5.3 ± 5.2	<0.001

* As determined by Student’s t-test or the chi-square test for recurrence vs. no recurrence. AAD, antiarrhythmic drug at AF recurrence; ABL, ablation; AF, atrial fibrillation; BMI, body mass index; BSA, body surface area; BUN, blood urea nitrogen; CHA_2_DS_2_, congestive heart failure, hypertension, age = 75 years, diabetes mellitus, stroke; DM, diabetes mellitus; Hb1Ac, glycated hemoglobin; HDL, high-density lipoprotein; HF, heart failure; LDL, low-density lipoprotein; TIA, transient ischemic attack; and VASc, vascular disease, age 65–75 years, sex category.

**Table 2 jcm-09-03164-t002:** Echocardiographic and ablation findings.

	Overall	No Recurrence Group	Recurrence Group	* *p* Value
	(*n* = 2221)	(*n* = 1490)	(*n* = 731)	
Echocardiographic findings				
LA dimension (mm)	41.2 ± 6.1	40.4 ± 5.9	42.8 ± 6.1	<0.001
LV ejection fraction (%)	55.1 ± 5.8	55.4 ± 5.6	54.4 ± 6.3	0.001
LA dimension ≥45 mm (%)	519 (23.6)	277 (18.8)	242 (33.3)	<0.001
Ablation findings				
Total ablation time (min)	114 ± 54	102 ± 47	129 ± 59	<0.001
Total fluoroscopy time (min)	68 ± 34	63 ± 32	76 ± 36	<0.001
Total procedure time (min)	275 ± 108	252 ± 98	309 ± 113	<0.001
Ablation lesion set				
CPVI, *n* (%)	2221 (100)	1490 (100)	731 (100)	NA
CTI, *n* (%)	1425 (64.0)	907 (61.0)	518 (71.0)	<0.001
CFAE, *n* (%)	507 (22.8)	288 (19.3)	219 (30.0)	<0.001
Linear ablation, *n* (%)	268 (12.1)	131 (8.8)	137 (18.7)	<0.001
Biatrial ablation, *n* (%)	520 (23.0)	236 (16.0)	284 (39.0)	<0.001

* As determined by Student’s t-test or the chi-square test for recurrence vs. no recurrence. CFAE, complex fractionated atrial electrogram; CPVI, circumferential pulmonary vein isolation; CTI, cavotricuspid isthmus; LA, left atrium; and LV, left ventricle.

**Table 3 jcm-09-03164-t003:** Clinical predictors of atrial fibrillation recurrence after catheter ablation.

	Univariate			Multivariate Model 1 *			Multivariate Model 2 †		
	HR	95% CI	*p* Value	HR	95% CI	*p* Value	HR	95% CI	*p* Value
Age	1.013	1.006–1.020	<0.001	1.007	1.000–1.015	0.051	1.008	1.000–1.015	0.043
Female	0.978	0.817–1.172	0.810	1.013	0.840–1.222	0.890	1.013	0.840–1.222	0.893
Non-paroxysmal AF	2.587	2.231–2.999	<0.001	2.254	1.919–2.648	<0.001	2.238	1.905–2.629	<0.001
Overweight or obese	1.216	1.006–1.470	0.043	1.391	1.176–1.937	0.018	1.314	1.107–1.559	0.020
HF	1.355	1.119–1.641	0.002	1.070	0.879–1.302	0.499	1.076	0.884–1.310	0.464
HTN	1.212	1.042–1.410	0.013	1.076	0.917–1.264	0.777	1.068	0.910–1.254	0.421
DM	1.325	1.032–1.700	0.027	1.231	0.955–1.588	0.109	1.240	0.962–1.599	0.097
LA dimension ≥45 mm	1.918	1.646–2.234	<0.001	1.279	1.005–1.033	0.004	1.284	1.085–1.518	0.004
MR LGE ≥25%	1.022	1.014–1.031	<0.001				1.726	1.330–2.239	<0.001
AF duration, per year	1.017	1.003–1.031	0.017	1.019	1.005–1.033	0.008	1.020	1.006–1.034	0.004

* Adjusted for age, sex, persistent AF (stepwise model (covariates: age, sex, non-paroxysmal AF, overweight, HF, HTN, DM, LA dimension, and AF duration)); † Adjusted for age, sex, persistent AF (stepwise model (covariates: age, sex, non-paroxysmal AF, overweight, HF, HTN, DM, LA dimension, AF duration, and MR LGE ≥25%)); AF, atrial fibrillation; CI, confidence interval; DM, diabetes mellitus; HF, heart failure; HTN, hypertension; LA, left atrium; and MR LGE, late gadolinium enhancement in cardiac magnetic resonance images.

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
