# Peer review of "Atrial Substrate Underlies the Recurrence after Catheter Ablation in Patients with Atrial Fibrillation"

_jcm, 2020, doi:10.3390/jcm9103164_

Round 1

Reviewer 1 Report

Atrial recurrence is a known issue occuring after PVI. In this article, the authors carefully characterized and stratified the clinical outcomes and predictors associated to repeated ablation, based on their clinical observations.

The recurrence appeared to be strongly associated with AF comorbidities, with non paroxysmal AF and with greater LGE.

Could the authors emphasize the novelty of their findings within the discussion part, since most of their observations were previously described elsewhere (PMID: 31933685, PMID: 2790952).

Could the authors be able to evaluate the LGE total area before and after the PVI in order to measure the impact of each individual ablations (e.g after 1 or 2 or 3 ablations...) on the fibrosis extent? Thus they could assess the impact on the total fibrosis level and evaluate the associated risk to develop further atrial arrhythmias.

Would the authors be able to discriminate an anatomical pattern (With LGE) that could be associated with an increased risk of recurrence (e.g Increased fibrosis areas close to PV) ? [Could the authors identify some anatomical patterns that could be associated with a triggered ectopic activity ?]

Author Response

Responses to Reviewer #1’s Comments

Reviewer #1: Atrial recurrence is a known issue occuring after PVI. In this article, the authors carefully characterized and stratified the clinical outcomes and predictors associated to repeated ablation, based on their clinical observations. The recurrence appeared to be strongly associated with AF comorbidities, with non-paroxysmal AF and with greater LGE.

Comments:

  1. Could the authors emphasize the novelty of their findings within the discussion part, since most of their observations were previously described elsewhere (PMID: 31933685, PMID: 2790952).?

Author’s response: We appreciate your helpful comment. AF ablation is associated with various recurrence rates, mostly due to patient-specific preprocedural factors and specific procedural factors.[1] Predictors of recurrence after AF ablation divided depending on the underling structural, electrical and autonomic substrate.[1,2] Our result showed that that catheter ablation is an effective strategy for rhythm control in patients with non-paroxysmal AF requiring multiple procedures and in those with paroxysmal AF and MR LGE can be quantified and visualized as predictors of recurrence after AF ablation. MR LGE to identify the atrial substrate is important in two aspects, and one is that it can avoid unnecessary AF ablation procedures, the other is that it can be feasible and effective for targeting re-entrant drivers or anatomic MRI LGE detected gaps in the AF ablation strategy.[3,4] In the present study, our result showed quantification that recurrence after ablation was significantly higher in MR LGE of 25% or more, and this is expected to be helpful in the ablation strategy in addition. To address your comment, we have added this in the Discussion section.

Manuscript change: “AF ablation is associated with various recurrence rates, mostly due to patient-specific preprocedural factors and specific procedural factors.[14] Predictors of recurrence after AF ablation divided depending on the underling structural, electrical and autonomic substrate.[14,15]”,“MR LGE to identify the atrial substrate is important in two aspects, and one is that it can avoid unnecessary AF ablation procedures, the other is that it can be feasible and effective for targeting re-entrant drivers or anatomic MRI LGE detected gaps in the AF ablation strategy.[23,24]”, “ In the longer term, efforts to identify atrial substrate using MR LGE would allow more patient-centered intervention and patient-tailored decision-making for better clinical outcome in patients who performed AF ablation.”(Page 11, line 293-295; line 324-331).

  1. Could the authors be able to evaluate the LGE total area before and after the PVI in order to measure the impact of each individual ablations (e.g after 1 or 2 or 3 ablations...) on the fibrosis extent? Thus they could assess the impact on the total fibrosis level and evaluate the associated risk to develop further atrial arrhythmias.

Author’s response: We appreciate your excellent comment. We fully agree with your comment. It has recently been reported that personalized computational models of the fibrotic atrial substrate derived from MR LGE before and after AF ablation can be used to non-invasively determine the strategy for AF ablation.[3] However, this study is longitudinal retrospective data. Our study is inherently limited by its nonrandomized, retrospective study. We did not perform MR LGE after the PVI to measure the impact of each individual ablations. We are planning a prospective study in the future to fully evaluate how the fibrotic substrate alters before and after AF catheter ablation. To address your comment, we have added this in the Limitation section.

Manuscript change: “Fourth, we did not perform MR LGE after the AF ablation to evaluate how the fibrotic substrate alters before and after the procedure.” (Page 12, line 337-338).

  1. Would the authors be able to discriminate an anatomical pattern (With LGE) that could be associated with an increased risk of recurrence (e.g Increased fibrosis areas close to PV) ? [Could the authors identify some anatomical patterns that could be associated with a triggered ectopic activity?

Author’s response: We appreciate your excellent comment. Unfortunately, we did not analyze because of limitations of our nonrandomized, retrospective study. Interestingly, as you point out, there was a retrospective study that PVI lesions close to ablation area in combination with unaffected fibrosis, indicate a pro-arrhythmic substrate that gives rise to new triggered activity.[3] These lesions sustain recurrent AF, sometimes in combination with remaining re-entrant drivers unaffected by ablation. Prospective studies need to be conducted to fully evaluate to identify anatomical patterns that could be associated with a triggered ectopic activity. 

References

  1. Garvanski, I.; Simova, I.; Angelkov, L.; Matveev, M. Predictors of Recurrence of AF in Patients After Radiofrequency Ablation. Eur Cardiol 2019, 14, 165-168, doi:10.15420/ecr.2019.30.2.
  2. Sultan, A.; Luker, J.; Andresen, D.; Kuck, K.H.; Hoffmann, E.; Brachmann, J.; Hochadel, M.; Willems, S.; Eckardt, L.; Lewalter, T., et al. Predictors of Atrial Fibrillation Recurrence after Catheter Ablation: Data from the German Ablation Registry. Sci Rep 2017, 7, 16678, doi:10.1038/s41598-017-16938-6.
  3. Ali, R.L.; Hakim, J.B.; Boyle, P.M.; Zahid, S.; Sivasambu, B.; Marine, J.E.; Calkins, H.; Trayanova, N.A.; Spragg, D.D. Arrhythmogenic propensity of the fibrotic substrate after atrial fibrillation ablation: a longitudinal study using magnetic resonance imaging-based atrial models. Cardiovasc Res 2019, 115, 1757-1765, doi:10.1093/cvr/cvz083.
  4. Fochler, F.; Yamaguchi, T.; Kheirkahan, M.; Kholmovski, E.G.; Morris, A.K.; Marrouche, N.F. Late Gadolinium Enhancement Magnetic Resonance Imaging Guided Treatment of Post-Atrial Fibrillation Ablation Recurrent Arrhythmia. Circ Arrhythm Electrophysiol 2019, 12, e007174, doi:10.1161/CIRCEP.119.007174.

Reviewer 2 Report

This is a useful clinical study.

My main concerns relate to use of statistics. I see you have used t test a lot. The problem here is you are comparing so many variables that the t statistic becomes meaningless. In fact with 20 variables you will be aproaching 1 100% chance of finding statistical significance that is false. You need to factor in some correction for the equivalent of multiple comparisons. This is a very common statistical fault. You my find that a lot of differences disappear.

Secondly you are using P values to characerise differences between groups. You have made P<0.05 your threshold for statistical significance (in Methods), but you have presented lots of exact values or less than something other than 0.05 values. This is not correct use of statistics. Please either explain what you mean by p=0.05-0.01, 0.01-0.005 etc, such as 'probably not real, maybe real, possible of interest' etc, or define your threshold for statistical significance (e.g. P<0.05) and stick to it. This is perhaps new for you but it is how the sector (biomed) is changing.

Author Response

Responses to Reviewer #2’s Comments

Reviewer #2: This is a useful clinical study. 

Comments:

  1. My main concerns relate to use of statistics. I see you have used t test a lot. The problem here is you are comparing so many variables that the t statistic becomes meaningless. In fact with 20 variables you will be aproaching 1 100% chance of finding statistical significance that is false. You need to factor in some correction for the equivalent of multiple comparisons. This is a very common statistical fault. You my find that a lot of differences disappear.

Author’s response: We appreciate your excellent comment. We fully agree with your comment. However, what we analyzed in baseline characteristics is to screen to select clinical relevant variables to identify predictors of clinical recurrence following ablation, not to show itself as the final outcome of the difference between the two groups using the p-value.

  1. Secondly you are using P values to characerise differences between groups. You have made P<0.05 your threshold for statistical significance (in Methods), but you have presented lots of exact values or less than something other than 0.05 values. This is not correct use of statistics. Please either explain what you mean by p=0.05-0.01, 0.01-0.005 etc, such as 'probably not real, maybe real, possible of interest' etc, or define your threshold for statistical significance (e.g. P<0.05) and stick to it. This is perhaps new for you but it is how the sector (biomed) is changing.

Author’s response: We appreciate your helpful comment. We know that the fact that p-value acceptance criteria are open to misuse and misinterpretation is a widely debated issue. To address your comment, we have defined our threshold for statistical significance in the manuscript.

Manuscript change: “P-values less than 0.05 were regarded as statistically noticeable in our study.” (Page 4, line 150-151)
